# Emotional Complications in Midwives Participating in Pregnancy Termination Procedures—Polish Experience

**DOI:** 10.3390/ijerph17082776

**Published:** 2020-04-17

**Authors:** Kornelia Zaręba, Jolanta Banasiewicz, Hanna Rozenek, Michał Ciebiera, Grzegorz Jakiel

**Affiliations:** 1First Department of Obstetrics and Gynecology, Center of Postgraduate Medical Education, 01-004 Warsaw, Poland; grzegorz.jakiel@o2.pl; 2Department of Medical Psychology and Medical Communication, Medical University of Warsaw, 00-575 Warsaw, Poland; jolantabanasiewicz.wum@gmail.com (J.B.); hanna.rozenek@wum.edu.pl (H.R.); 3Second Department of Obstetrics and Gynecology, Center of Postgraduate Medical Education, 01-809 Warsaw, Poland; michal.ciebiera@cmkp.pl

**Keywords:** abortion, termination of pregnancy, midwife, burnout, occupational burnout, stress, workplace stress

## Abstract

*Background*: Ethically controversial medical procedures, such as the termination of pregnancy, are frequently associated with a discrepancy between personal attitude and values versus requirements related to a professional situation. The study aimed to assess emotional complications in midwives participating in pregnancy termination procedures. *Methods:* The study included 181 midwives working in state-governed healthcare facilities in central and eastern Poland. The Oldenburg Burnout Inventory (OLBI) and the present authors’ own questionnaire were used in the study. The results indicating the level of occupational burnout were presented in two scales: the exhaustion scale and the disengagement scale. *Results:* The study revealed that 48% of midwives had never participated in pregnancy termination procedures due to fetal defects. The level of occupational burnout described with the exhaustion factor (t = 2.06; *p* < 0.041) and disengagement factor (t = 2.96; *p* < 0.003) was significantly higher in the group of midwives participating in pregnancy termination procedures due to fetal defects than in the group of midwives who did not participate in pregnancy terminations. The most common factors contributing to burnout reported by midwives who participated in pregnancy terminations were: moral dilemmas (68%), seeing the aborted fetus (65%), anticipating the child’s death in case it was born with signs of life (59%) and the lack of professional psychological support for medical personnel (56%). *Conclusions:* Importantly, pregnancy termination should be performed by persons who find such procedures acceptable from the viewpoint of their value system. It is a protective factor in regards to work with women who undergo terminations. Moreover, developing a system of informational and psychological support for midwives participating in pregnancy termination procedures is also a significant aspect.

## 1. Introduction

Professional literature is devoid of the topic of feelings and possible emotional complications regarding medical personnel participating in pregnancy termination procedures. Ethically controversial medical procedures are frequently associated with a discrepancy between personal attitude and values versus requirements related to a professional situation. Cognitive dissonance is associated with experiencing unpleasant psychological stress induced by undertaking actions inconsistent with personal beliefs, values and expressed views [1]. When deciding about the choice of profession, midwives had planned to assist in childbirth, take care of the mother and child and share the happiness of the miracle of birth. Pregnancy termination is connected with undertaking actions which are supposed to lead to fetal death in a pregnant woman. Complying with doctors’ orders and passive anticipation of the child’s death after delivery is difficult to accept by some midwives [2]. At the same time, many of them agree to participate in such procedures because of the fear of the disapproval of supervisors or losing a job. Researchers from the Medical University in Lublin indicated that 84% of midwives had never invoked the so-called “conscience clause”, although they performed actions which they described as inconsistent with their conscience [2].

Medically, pregnancy termination may be divided into pharmacological and surgical termination [3], while legally it means “artificial induction of the removal of the embryo or the fetus resulting in its death” [4]. Artificial abortion may be performed legally in accordance with regulations obligatory in a specific country. It may be performed due to medical, eugenic, legal or social indications.

Legal regulations concerning pregnancy termination in Poland are specified in the Family Planning, Human Embryo Protection and Conditions of Permissibility of Abortion Act of 7 January, 1993. According to Polish law, it is acceptable to terminate a pregnancy following the permission of the doctor and the pregnant woman in three instances: the pregnancy poses a direct threat to maternal health or life;prenatal screening or other medical evidence indicate a high probability of severe and irreversible fetal anomaly or incurable life-threatening disease;there is a reason to believe that pregnancy is the result of an unlawful act [5].

The second of the above-mentioned situations is the most common cause of pregnancy terminations in Poland. In such a case, termination may be performed until the moment when the fetus is able to live independently outside of the mother’s body. The majority of authors specified that according to Polish law the fetal age should not exceed six months [6].

According to official data, 1057 legal terminations were recorded in Poland in 2017, including 22 procedures performed due to a threat to the mother’s life and health, 1035 due to severe and irreversible fetal defects and 0 due to criminal offenses (Report of the Council of Ministers, 2019) [7].

Depending on the gestational age, artificial abortion may be induced with instrumental and pharmacological methods. The majority of pregnancy terminations due to fetal defects are performed during the second trimester of pregnancy because of a relatively long period of waiting for the results of prenatal tests. In Poland, during that period of pregnancy, pharmacological methods are mainly used with oral or intravaginal administration of substances inducing uterine contractions. In regards to late gestation, the termination procedure becomes similar to natural delivery. It is characterized by considerable duration, as it may last up to several or even several dozen hours. During such procedures, the child may be relatively big and weigh even 500 g. The procedure ends with the curettage of the uterine cavity. In the majority of cases of induced abortion, the fetus is born non-viable. Substances administered to the mother to induce delivery and strong uterine contractions cause the fetus to die. Such a situation is less stressful for midwives. After delivery the cadaver is prepared to be transported for histopathological exam. A lot more emotions arise in a situation of a live birth, especially in case of a so-called late termination. The midwife is expected to wait passively until the vital functions of the child stop. The awareness of being somehow responsible for the situation and the knowledge that nothing can be done for the dying child apart from providing thermal comfort causes anger, feelings of helplessness and depression in many midwives [1,8]. It is exceptionally hard for the mother and her partner, but also for the midwife who takes care of the woman. 

Parents who decided to terminate a pregnancy experience sadness, anger and guilt, especially if they are already emotionally bonded with the child, or chose the child’s name. They attempt at explaining their choice by the child’s wellbeing and expect understanding and absolution of their decisions by the medical personnel [9]. Some people experience disappointment and frustration because of the low effectiveness of contraction-inducing agents, prolonged delivery and pain. It is not uncommon that negative emotions cause aggressive behavior to midwives and physicians. Midwives, for whom just the participation in such a delivery is stressful, are expected to deal with the emotions of the patient, their partner and sometimes even other family members [9]. Numerous factors make the participation in an induced abortion a traumatic experience. Studies show that participation in multiple pregnancy termination procedures increased the probability of developing posttraumatic stress disorder [10]. 

The necessity of psychological support has long been acknowledged in regards to the personnel of hospices and oncological and psychiatric departments. In order to release emotional tension, midwives most commonly share their feelings within their professional team or tackle the issue when talking to their family members. A new regulation of the Polish Minister of Health became obligatory on 1 January, 2019. According to its item, no. XV.11, an employer of midwives, nurses or physicians working with patients who experience a serious disease or loss of a child is obliged to provide psychological support in coping with stress [11]. Regrettably, this regulation is extremely rarely put into practice. Few midwives use the help of psychologists, psychiatrists or clergymen. It is probably associated with low availability of such services [8]. 

The conscience clause is a specific legal clause which gives medical personnel the possibility of refusal in performing procedures which are incompliant with one’s ethical values. In Poland this clause may be invoked by physicians, nurses and midwives. According to the Polish “Act on the Professions of Nurse and Midwife” dated from 15 July, 2011 “a nurse and a midwife may refrain, after prior notifying the supervisor in writing, from performing healthcare services which are inconsistent with her ethical values, excluding cases of serious danger to the patient’s health or possibility of death” [12]. However, this regulation is not observed in numerous healthcare facilities, especially in regards to nurses and midwives, who do not attempt at exercising their rights in this matter because of fear of losing the job. The supervisors in centers where personnel rights are observed are faced with the dilemma of how to accommodate women’s right to terminate a pregnancy and the right of medical personnel to refuse participation in such procedures [13].

Job satisfaction is the feeling of well-performed responsibilities and certainty that the time and effort put into work was not wasted [14]. Such feelings are rarely experienced by midwives participating in pregnancy termination procedures. A large group reported the feeling of meaninglessness of undertaken actions and moral dilemmas [15]. Some midwives justified their participation in pregnancy termination procedures as the source of job satisfaction by the fact of preventing possible suffering of the child and the parents [8].

The problem of occupational burnout in medical personnel was analyzed by many authors [13]. According to German researchers, approximately 69% of people experience occupational burnout in their lifetime [16]. The authors demonstrated that the problem was present in numerous representatives of medical professions, sometimes even before they started their professional career. Researchers indicated that the problem occurred in as many as 31% of medical students [17], approximately 45% of gynecologists [18] and approximately 20%–54% of midwives [19]. Systematic research into this phenomenon was started in the 1970s by two independent authors—Freudenberger and Maslach. In 1974, Freudenberger published an article entitled “Staff burn-out” where he included observations related to burnout in the work of volunteers who were helping at a center for drug addicts. Subsequently, in 1980, he expanded his view in a book entitled “The High Cost of High Achievement”. He provided a description of exhaustion, tiredness and frustration with the job which occurred in individuals whose profession involved helping other people [20]. Maslach conducted a study on the ways of coping with emotions at workplace. She noted that chronic tension, which is experienced by individuals professionally helping others, may lead to emotional exhaustion, reduced commitment to work and cynicism. She presented the results in an article entitled “Burned out” published in 1976. Subsequently, she expanded her views and presented a three-factor concept of occupational burnout with emotional exhaustion, depersonalization and reduced personal accomplishment being the dimensions of the phenomenon [21]. According to this concept, emotional exhaustion refers to a feeling of excessive mental load, inner emptiness and the lack of physical and mental energy to act. Such symptoms of burnout are commonly associated with various psychosomatic problems such as depression and anxiety [22,23]. Reduced personal accomplishment presents as a negative assessment of one’s professional competence, a sense of misunderstanding and a lack of confidence in one’s ability to solve problems occurring at work [20,24,25,26]. At the end of the 1990s Demerouti presented a new perspective regarding the phenomenon of occupational burnout [27]. The researchers presented Job Demands-Resources model (JD-R model) according to which occupational burnout occurs due to requirements of the work environment associated with insufficient resources. The authors listed poor organization of work, role conflicts, no sense of control and excessive workload as demands. JD-R model resources include factors associated with work environment and personality ones. The resources in the described model encompass: the permanence of employment, satisfactory remuneration, appropriate conditions at work, friendly atmosphere, support from co-workers and supervisors, clearly defined scope of responsibilities, having influence on the situation at work and the sense of accomplishment. The model describes two components making up occupational burnout: exhaustion and disengagement from work. Exhaustion means the feeling of inner emptiness, tiredness, extreme mental and physical weakness. Disengagement is defined as being distanced from professional responsibilities, aims, values observed at work, towards one’s subordinates, co-workers and supervisors [28].

## 2. Materials and Methods 

The study aimed to assess emotional complications in midwives participating in pregnancy termination procedures. It was designed as a cross-sectional prospective study and conducted between 1 April, 2019 and 31 December, 2019. It was approved by the Bioethics Committee [REDACTED]. 

Midwives working in state-funded hospitals (they are the only facilities in which pregnancy terminations are performed in Poland) of middle-eastern Poland were randomly asked to complete an anonymous questionnaire and the Oldenburg Burnout Inventory (OLBI) questionnaire. The study was conducted in 5 hospitals and the respondents were randomly selected. 

Study group inclusion criteria:Midwives with the job seniority of at least 1 year.Employment at state-funded healthcare facilities.Consent to participate in the study.

Study group exclusion criteria: Midwives who are not professionally active or with the job seniority of less than a year.Midwives working at healthcare facilities which are not state funded.No consent to participate in the study.

The researcher asked the respondents to complete questionnaires. Completing one questionnaire lasted around 1 h. A total of 300 midwives were invited to participate in the study. A total of 200 of them consented to participate and 181 returned correctly completed questionnaires. A 100% return rate was achieved with 90.5% of questionnaires being completed correctly.

The questionnaire developed for the needs of the study included questions concerning demographic data: age, place of residence, level of education, marital status, number of children, job seniority, participation in termination procedures. The OLBI questionnaire was used in the study. Information concerning the sociodemographic and professional situation of study group midwives was collected with the present authors’ own questionnaire. The OLBI questionnaire adapted for Polish studies consists of 16 questions. The respondents have to select an answer on a 4-grade scale (Appendix A). The results reflect the level of occupational burnout in two scales: the scale of exhaustion and disengagement. It includes questions formulated in a positive and negative way and is prepared for workers of the general population (a variety of professions). The Polish language version of the OLBI was developed by Cieślak following the consent of the author of the original version of the questionnaire. The psychometric properties were verified by Baka, Basińska [25]. Exhaustion is defined as a result of long-lasting physical, mental and emotional tension which is due to excessive workload and stressors at workplace. Disengagement from work is perceived as the lack of interest in performed activities, performing tasks mechanically, without enthusiasm, or aversion to work [25]. The reliability of the English version of the OLBI was assessed on the basis on Cronbach-α internal consistency at 0.74–0.87. The stability of the questionnaire: moderate level of correlation coefficients (r = 0.51, *p* < 0.001, exhaustion scale; r = 0.34; *p* < 0.01, disengagement scale). 

Prior to conducting the final analyses, preliminary analyses were performed which included descriptive statistics: frequency expressed as numbers and percentages, means, standard deviations. Kolmogorov–Smirnov test was used to test the normality of distributions of the obtained results. The final analyses were based on the Student’s t-test for independent variables to compare groups of midwives who participated or not in pregnancy termination procedures. The homogeneity of variance was checked with Levene’s test prior to performing the Student’s *t*-test. The criterion of statistical inference was set at the level of significance of *p* < 0.05. Analyses were conducted with SPSS 10.0 statistical package (IBM Corp. Released 2010, SPSS Statistics for Windows Version 19.0, Armonk, NY, USA).

## 3. Results

The average age in the group was 40.79 years (SD = 8.55), with the largest percentage of women aged 36–55 (82%). The largest group of respondents (46%) lived in big cities with over 200,000 residents. Midwives living in towns (41%) and in villages (13%) also participated in the study. The largest study group midwives (75; 42%) completed secondary education. This group includes those who attended a Medical Vocational School for midwives. This form of educating midwives functioned until 1999. Bachelor’s degree studies were completed by 33% of the respondents, while Master’s degree studies were completed by 25%. The majority of respondents were married (68%). The study group also included single (never married) women (19%), divorcees (7%), persons living in an informal relationship (4%) and widows (2%). The majority of study group midwives (80%) had children and as regards the remaining 20%—a total of 16% of the whole study group would like to have children.

A total of 95.5% of study group women declared Roman Catholic denomination, while regular religious practice was declared by 45% of the respondents. One person belonged to a neopagan Wicca movement. No religious belief was declared by 1.5% of the study group. The data are presented in Table 1.

### 3.1. Professional Characteristics of the Study Group

The average length of seniority in the study group of midwives was 16.73 years (SD = 8.63). A total of 48% of study group midwives had never participated in pregnancy termination procedures due to fetal defects (Table 2). In the majority of cases, this was due to the fact that no such procedures were performed at their workplace (43.3%), while the remaining 4.7% refused to participate in such procedures. In regards to the 52% of respondents who did participate in pregnancy terminations, the largest group expressed concerns about the consequences of refusal (22.1%) or they agreed with the decision of the patients (14.9%). The remaining study group midwives agreed to participate as they had not realized the consequences of such actions (8.3%). Some respondents did not answer this question.

### 3.2. Factors Contributing to Burnout in Pregnancy Termination Procedures

The number of procedures in which midwives participated constituted an important risk factor of emotional complications. Table 2 presents the distribution of the number of midwives in relation to the frequency of participation in pregnancy termination procedures. 

The most common factors contributing to burnout reported by midwives who participated in pregnancy terminations were moral doubts (Table 3). They were reported by 68% of midwives. The second most common factor was seeing the aborted fetus. The respondents also found it very stressful to anticipate the death of the child if there were any signs of life. Numerous midwives did not have access to professional psychological support addressed to medical staff. Other difficulties reported by midwives included refraining from expressing their feelings and views, and the necessity to participate in multiple procedures of this kind. Midwives were also stressed by the fact that they did not possess sufficient knowledge on working with persons who decided to terminate a pregnancy.

When completing the space “Others”, the midwives who participated in pregnancy terminations had the possibility of listing factors which were the most stressful in pregnancy terminations. They listed the following stressors: irritating lack of knowledge about contraception in patients;indifference of patients;conflict with religious beliefs;strange behaviors of patients;lack of psychological support for patients;patients’ unawareness of what pregnancy termination is associated with;cleaning after the procedure;placing dead fetuses into containers with formalin.

### 3.3. Participation in Pregnancy Termination Procedures and Occupational Burnout

The comparison of groups of midwives who participated and did not participate in pregnancy termination procedures due to fetal defects demonstrated statistically significant differences concerning two factors in the OLBI scale which was used to measure occupational burnout. The level of occupational burnout described with the exhaustion factor (t = 2.06; *p* < 0.041) and disengagement factor (t = 2.96; *p* < 0.003) was significantly higher in the group of midwives participating in pregnancy termination procedures due to fetal defects than in the group of midwives who did not participate in pregnancy terminations. The results of the analysis performed with the Student’s *t*-test are presented in Table 4.

The comparison of groups of midwives who participated and did not participate in pregnancy termination procedures due to fetal defects demonstrated statistically significant differences concerning two factors in the OLBI scale, which was used to measure occupational burnout. It demonstrated statistically significant differences regarding one factor in the OLBI scale which measures occupational burnout: the level of occupational burnout described with the disengagement factor (t = 3.46; *p* < 0.001) was significantly higher in the group of midwives participating in pregnancy termination procedures than in the group of midwives who did not participate in pregnancy terminations.

In regards to the group of midwives participating in pregnancy termination procedures, Pearson correlation (r) analysis showed no statistically significant relationship between the indices of occupational burnout in the OLBI scale and the number of terminations in which the midwives participated.

## 4. Discussion

Nursing professionals obtain lower scores in the mental component of HRQoL (Health-Related Quality of Life) due to work-related stress and repeated contact with situations of suffering [29]. The analysis of results revealed no statistically significant correlations between the level of occupational burnout and the majority of sociodemographic variables. No statistically significant relationship was found between occupational burnout and age, marital status, job seniority, place of residence and religious beliefs. The results of studies conducted by other authors are highly diversified in this matter [30,31]. Baran and Piątek reported a correlation between burnout and the age and job seniority in midwives [32]. Other studies conducted with Polish nurses indicated the presence of a correlation between occupational burnout and age. However, no relationship was revealed between occupational burnout and job seniority [33]. Uchmanowicz et al. revealed that the most significant factor was emotional exhaustion [34]. A study by Owczarek et al. conducted with nurses of selected hospital departments confirmed a relationship between job seniority and occupational burnout [35]. The analysis of results concerning occupational burnout in midwives in the context of the participation in pregnancy termination procedures showed that the participation in abortions may contribute to negative psychological consequences. Statistical analyses demonstrated that the level of occupational burnout was higher in midwives participating in pregnancy termination procedures than in those who did not participate in terminations. Similar results were presented in a study assessing the relationship between traumatic events associated with the work of midwives and occupational burnout [30]. Occupational burnout was assessed with MBI questionnaire (Maslach Burnout Inventory questionnaire) in that study. The classification of 13 traumatic events experienced by midwives also comprised pregnancy termination procedures. A total of 23 midwives were examined who participated in induced abortion procedures. A moderate positive correlation was found between emotional exhaustion and participation in pregnancy terminations. Moreover, a positive correlation was revealed between the depersonalization dimension and participation in terminations [32]. A cross-sectional correlation research study conducted in 282 midwives working in labor wards of hospitals in Iran revealed that high scores in emotional exhaustion and low scores in personal accomplishment predisposed to high levels of job burnout [31].

Surprisingly, when conducting the present study no statistically significant correlation was found between the level of occupational burnout and the number of pregnancy termination procedures in which the midwives participated. The result may be associated with the phenomenon of “frozen emotions”. It is a specific protective barrier against negative emotions such as fear, or guilt which may appear in such situations. Different results were obtained at Kanazawa University in Japan. A study published at the beginning of 2013 showed that the risk of occupational burnout increased with an increased number of abortions in which a midwife participated [36].

Moral and ethical dilemmas, work inconsistent with one’s system of values largely contribute to the development of occupational burnout. According to Maslach “(…) People do their best if they believe in what they do, when they can feel pride, integrity and respect for oneself” [37]. If the work is inconsistent with their system of values, burnout occurs more rapidly. Some midwives may even experience identity crisis if the work is inconsistent with their accepted axioms. Researchers from the Republic of South Africa emphasized that midwives who participated in pregnancy termination procedures out of their own will, consistently with their views (and not because of reasons like fear of losing a job), experienced considerably fewer negative emotions associated with pregnancy termination procedures [38]. The risk of occupational burnout increases with stronger feelings of meaninglessness of undertaken actions. Surprisingly, the study of Polish midwives showed no correlation between the level of commitment to religious beliefs and occupational burnout. It may be assumed that in this case the real hierarchy of values, and not the denomination, constituted the differentiating factor (the majority of the respondents were Catholic).

The selection of a group of midwives exposed to occupational burnout with standardized psychological questionnaires may facilitate the implementation of suitable protective management including: meditation, progressive relaxation, breathing exercises, mindfulness training. A systematic review analysis demonstrated that mindfulness training reduced the levels of burnout [39,40].

### 4.1. Limitations of the Study

The problem tackled in the present study is deemed controversial. Undoubtedly, it might influence the number of respondents, and the subsequent representativeness of the obtained results. The same reasons may be provided for a possible overestimation of the respondents’ opinions concerning data (like the number of procedures in which they participated) when trying to emphasize the difficulties they faced.

### 4.2. Strengths of the Study

Few publications described the moral and psychological problems associated with pregnancy termination procedures from the viewpoint of participating midwives. The present study specified the strongest stressors in the work of midwives participating in pregnancy termination procedures. It is also the first study conducted in such an extensive group of midwives participating in the procedures in Poland where the issue of abortion is still highly controversial and is a topic of political debate.

## 5. Conclusions

The obtained results showed that midwives experienced numerous negative consequences associated with the participation in pregnancy termination procedures. A higher level of occupational burnout was observed in the group of midwives who participated in pregnancy termination procedures compared to midwives who did not participate. Importantly, work with women who undergo pregnancy termination should be performed by individuals who find such procedures acceptable from the viewpoint of their value systems. The results also indicated the necessity to organize appropriate informational and psychological support groups for midwives participating in pregnancy termination procedures.

## Figures and Tables

**Table 1 ijerph-17-02776-t001:** General description of the study group.

Variable		*n* = 181
Age		40.79 ± 8.55
Sex	Women	181 (100%)
Size of the town/city	Village	23 (13%)
	<50,000	40 (22%)
	50,000–200,000	35 (19%)
	>200,000	83 (46%)
Religion	Catholicism	173 (95.5%)
	Orthodoxy	4 (2%)
	Baptism	1 (0.5%)
	Other	1 (0.5%)
	Agnostic	3 (1.5%)
Marital status	Unmarried	34 (19%)
	Married	123 (68%)
	Cohabitation	7 (4%)
	Divorced	12 (7%)
	Widow	4 (2%)
Children	No	36 (20%)
	Yes	145 (80%)
Job seniority in gynecology	1–5 years	24 (13%)
	6–10 years	21(12%)
	11–15 years	32 (18%)
	16–20 years	37 (20%)
	21–25 years	43 (23%)
	26–30 years	12 (7%)
	>30 years	12 (7%)
Education	Secondary	75 (42%)
	Bachelor’s degree	30 (33%)
	Master’s degree	46 (25%)

**Table 2 ijerph-17-02776-t002:** The number of midwives versus the frequency of participation in pregnancy termination procedures.

The Number of Midwife-Attended Procedures	*%* of Midwives
no participation	48
1 to 9	7.7
10 to 49	15.5
50 to 99	10
100 to 149	10
150 to 199	0.5
200 to 249	3.3
250 to 299	0
over 300	5

**Table 3 ijerph-17-02776-t003:** Factors contributing to burnout as reported by the respondents participating in pregnancy termination procedures.

Stressors in Pregnancy Termination	% of Midwives
Participation in multiple procedures	37
Seeing aborted fetuses	65
Waiting for the death of the child if born with signs of life	59
Working with persons experiencing negative emotions	30
No knowledge on working with persons who decided to terminate a pregnancy	21
Lack of professional psychological support for medical personnel	56
Moral dilemmas	68
No right to express one’s feelings and views	40
Others	10

**Table 4 ijerph-17-02776-t004:** Comparison of mean values of occupational burnout of midwives participating and not participating in pregnancy termination procedures due to fetal defects.

Occupational Burnout	Participation in Terminations Due to Fetal Defects	*n*	Mean	Standard Deviation	*t*	*p*
Exhaustion	No	99	18.86	3.58	−2.06	0.041
	Yes	81	19.89	3.01		
Disengagement	No	99	16.39	4.50	−2.96	0.003

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
