# Peer review of "Emotional Complications in Midwives Participating in Pregnancy Termination Procedures—Polish Experience"

_ijerph, 2020, doi:10.3390/ijerph17082776_

Round 1
Reviewer 1 Report
Dear Authors,
Your paper is interesting but I have doubts about methodological design. I would like you to resolve the following issues:
INTRODUCTION
The introduction is very long. It would be interesting to know epidemiological data.
METHODS
Authors must specify the research design.
What was the target population? How was the sample chosen? Authors must specify it.
Also, authors must specify sociodemographic variables.
DISCUSSION
The discussion is based on very old references. Authors should review more recent research.
REFERENCES
Many bibliographies are obsolete. The bibliographic citations used are more than 5 years old (85,7%). The authors must update and arrange the bibliography.
There are recent meta-analytic articles on risk factors in midwives and on burnout in midwives. It is recommended to cite any.
Author Response
Submited, please see the attachment

Reviewer 2 Report
It is a very interesting study to read and further research is highly expected. However, I just suggest one thing to be added. Like around 192, This study used OLBL and adapted for 16 questions in the questionnaire. To help reader's understanding better, it is suggested that presenting the consisted items of 16 questions in detail.
Author Response
Uploaded

Round 2
Reviewer 1 Report
Dear Authors,
His work still has deficiencies that have not been corrected:
METHODS
Authors must specify the research design. It´s not included and has been requested.
REFERENCES
Many bibliographies are obsolete. The bibliographic citations used are more than 5 years old (60,5%). The authors must update and arrange the bibliography.
There are recent meta-analytic articles on risk factors in midwives and on burnout in midwives:
- Suleiman-Martos N, Gomez-Urquiza JL, Aguayo-Estremera R, Cañadas-De La Fuente GA, De La Fuente-Solana EI, Albendín-García L. The effect of mindfulness training on burnout syndrome in nursing: A systematic review and meta-analysis. J Adv Nurs. 2020; 76(5):1124-1140. doi: 10.1111/jan.14318.
- Janighorban M, Dadkhahtehrani T, Najimi A, Hafezi S. The Correlation between Psychological Empowerment and Job Burnout in Midwives Working in the Labor Ward of Hospitals. Iran J Nurs Midwifery Res. 2020; 25(2):128-133. doi: 4103/ijnmr.IJNMR_100_19.
- ….
Author Response
Enclosed
